# Efficient Inference of Flexible Interaction in Spiking-neuron Networks

**Feng Zhou**[†]**, Yixuan Zhang**[‡]**, Jun Zhu**[†*]

[†]Dept. of Comp. Sci. & Tech., BNRist Center, THU-Bosch Joint ML Center, Tsinghua University
[‡]Data Science Institute, University of Technology Sydney
{zhoufeng6288, dcszj}@tsinghua.edu.cn, yixuan.zhang@uts.edu.au

## Abstract

Hawkes process provides an effective statistical framework for analyzing the time-dependent interaction of neuronal spiking activities. Although utilized in many real applications, the classic Hawkes process is incapable of modelling inhibitory interactions among neurons. Instead, the nonlinear Hawkes process allows for a more flexible influence pattern with excitatory or inhibitory interactions. In this paper, three sets of auxiliary latent variables (Pólya-Gamma variables, latent marked Poisson processes and sparsity variables) are augmented to make functional connection weights in a Gaussian form, which allows for a simple iterative algorithm with analytical updates. As a result, an efficient expectation-maximization (EM) algorithm is derived to obtain the maximum a posteriori (MAP) estimate. We demonstrate the accuracy and efficiency performance of our algorithm on synthetic and real data. For real neural recordings, we show our algorithm can estimate the temporal dynamics of interaction and reveal the interpretable functional connectivity underlying neural spike trains.

## 1 Introduction

One of the most important tracks in neuroscience is to examine the neuronal activity in the cerebral cortex under varying experimental conditions. Recordings of neuronal activity are represented through a series of action potentials or spike trains. The transmitted information and functional connection between neurons are considered to be primarily represented by spike trains (Kass et al., 2014; Kass & Ventura, 2001; Brown et al., 2004; 2002). A spike train is a sequence of recorded times at which a neuron fires an action potential and each spike may be considered to be a timestamp. Spikes occur irregularly both within and across multiple trials, so it is reasonable to consider a spike train as a point process with the instantaneous firing rate being the intensity function of point processes (Perkel et al., 1967; Paninski, 2004; Eden et al., 2004). An example of spike trains for multiple neurons is shown in Fig. 2a in the real data experiment.

Despite many existing applications, the classic point process models, e.g., Poisson processes, neglect the time-dependent interaction within one neuron and between multiple neurons, so fail to capture the complex temporal dynamics of a neural population. In contrast, Hawkes process is one type of point processes which is able to model the *self-exciting* interaction between past and future events. Existing applications cover a wide range of domains including seismology (Ogata, 1998; 1999), criminology (Mohler et al., 2011; Lewis et al., 2012), financial engineering (Bacry et al., 2015; Filimonov & Sornette, 2015) and epidemics (Saichev & Sornette, 2011; Rizoiu et al., 2018). Unfortunately, due to the linearly additive intensity, the vanilla Hawkes process can only represent the purely excitatory interaction because a negative firing rate may exist with inhibitory interaction. This makes the vanilla version inappropriate in the neuroscience domain where the influence between neurons is a mixture of excitation and inhibition (Maffei et al., 2004; Mongillo et al., 2018).

In order to reconcile Hawkes process with inhibition, various nonlinear Hawkes process variants are proposed to allow for both excitatory and inhibitory interactions. The core point of nonlinear Hawkes process is a nonlinearity which maps the convolution of the spike train with a causal influential kernel to a nonnegative conditional intensity, such as rectifier (Reynaud-Bouret et al., 2013),

---

[*]Corresponding author.

exponential (Gerhard et al., 2017) and sigmoid (Linderman, 2016; Apostolopoulou et al., 2019). The sigmoid mapping function has the advantage that the Pólya-Gamma augmentation scheme can be utilized to convert the likelihood into a Gaussian form, which makes the inference tractable. In Linderman (2016), a discrete-time model is proposed to convert the likelihood from a Poisson process to a Poisson distribution. Then Pólya-Gamma random variables are augmented on discrete observations to propose a Gibbs sampler. This method is further extended to a continuous-time regime in Apostolopoulou et al. (2019) by augmenting thinned points and Pólya-Gamma random variables to propose a Gibbs sampler. However, the influence function is limited to be purely exciting or inhibitive exponential decay. Besides, due to the nonconjugacy of the excitation parameter of exponential decay influence function, a Metropolis-Hastings sampling step has to be embedded into the Gibbs sampler making the Markov chain Monte Carlo (MCMC) algorithm further inefficient.

To address the *parametric* and *inefficient* problems in aforementioned existing works, we develop a flexible sigmoid nonlinear multivariate Hawkes processes (SNMHP) model in the continuous-time regime, **(1)** which can represent the *flexible excitation-inhibition-mixture* temporal dynamics among the neural population, **(2)** with the *efficient conjugate* inference. An EM inference algorithm is proposed to fit neural spike trains. Inspired by Donner & Opper (2017; 2018), three auxiliary latent variable sets: Pólya-Gamma variables, latent marked Poisson processes and sparsity variables are augmented to make functional connection weights in a Gaussian form. As a result, the EM algorithm has analytical updates with drastically improved efficiency. As shown in experiments, it is even more efficient than the maximum likelihood estimation (MLE) for the parametric Hawkes process in high dimensional cases.

## 2 OUR MODEL

Neurons communicate with each other by action potentials (spikes) and chemical neurotransmitters. A spike causes the pre-synaptic neuron to release a chemical neurotransmitter that induces impulse responses, either exciting or inhibiting the post-synaptic neuron from firing its own spikes. The addition of excitatory and inhibitory influence to a neuron determines whether a spike will occur. At the same time, the impulse response characterizes the temporal dynamics of the exciting or inhibiting influence which can be complex and flexible (Purves et al., 2014; Squire et al., 2012; Bassett & Sporns, 2017). Arguably, the flexible nonlinear multivariate Hawkes processes are a suitable choice for representing the temporal dynamics of mutually excitatory or inhibitory interactions and functional connectivity of neuron networks.

### 2.1 MULTIVARIATE HAWKES PROCESSES

The vanilla multivariate Hawkes processes (Hawkes, 1971) are sequences of timestamps $D = \{\{t_n^i\}_{n=1}^{N_i}\}_{i=1}^{M} \in [0, T]$ where $t_n^i$ is the timestamp of $n$-th event on $i$-th dimension with $N_i$ being the number of points on $i$-th dimension, $M$ the number of dimensions, $T$ the observation window. The $i$-th dimensional conditional intensity, the probability of an event occurring on $i$-th dimension in $[t, t + dt)$ given all dimensional history before $t$, is designed in a linear superposition form:

$$\lambda_i(t) = \mu_i + \sum_{j=1}^{M} \sum_{t_n^j < t} \phi_{ij}(t - t_n^j), \tag{1}$$

where $\mu_i > 0$ is the baseline rate of $i$-th dimension and $\phi_{ij}(\cdot) \geq 0$ is the causal influence function (impulse response) from $j$-th dimension to $i$-th dimension which is normally a parameterized function, e.g., exponential decay. The summation explains the self- and mutual-excitation phenomenon, i.e., the occurrence of previous events increases the intensity of events in the future. Unfortunately, one blemish is the vanilla multivariate Hawkes processes allow only nonnegative (excitatory) influence functions because negative (inhibitory) influence functions may yield a negative intensity which is meaningless. To reconcile the vanilla version with inhibitory effect and flexible influence function, we propose the SNMHP.

## 2.2 Sigmoid Nonlinear Multivariate Hawkes Processes

Similar to the classic nonlinear multivariate Hawkes processes (Brémaud & Massoulié, 1996), the $i$-th dimensional conditional intensity of SNMHP is defined as

$$\lambda_i(t) = \overline{\lambda}_i \sigma(h_i(t)), \quad h_i(t) = \mu_i + \sum_{j=1}^{M} \sum_{t_n^j < t} \phi_{ij}(t - t_n^j), \tag{2}$$

where $\mu_i$ is the base activation of neuron $i$, $h_i(t)$ is a real-valued activation and $\sigma(\cdot)$ is the logistic (sigmoid) function which maps the activation into a positive real value in $(0, 1)$ with $\overline{\lambda}_i$ being a upper-bound to scale it to $(0, \overline{\lambda}_i)$. The sigmoid function is chosen because as seen later, the Pólya-Gamma augmentation scheme can be utilized to make the inference tractable. After incorporating the nonlinearity, it is straightforward to see the influence functions, $\phi_{ij}(\cdot)$, can be positive or negative. If $\phi_{ij}(\cdot)$ is negative, the superposition of $\phi_{ij}(\cdot)$ will lead to a negative activation $h_i(t)$ that renders the intensity to 0; instead, the intensity tends to $\overline{\lambda}_i$ with a positive $\phi_{ij}(\cdot)$.

To achieve a flexible impulse response, the influence function is assumed to be a weighted sum of basis functions

$$\phi_{ij}(\cdot) = \sum_{b=1}^{B} w_{ijb} \tilde{\phi}_b(\cdot), \tag{3}$$

where $\{\tilde{\phi}_b\}_{b=1}^{B}$ are predefined basis functions and $w_{ijb}$ is the weight capturing the influence from $j$-th dimension to $i$-th dimension by $b$-th basis function with positive indicating excitation and negative indicating inhibition. The basis functions are nonnegative functions capturing the temporal dynamics of the interaction. Although basis functions can be in any form, in order for the weights to represent functional connection strength, basis functions are chosen to be probability densities with compact support that means they have bounded support $[0, T_\phi]$ and the integral is one. As a result, the $i$-th dimensional activation is

$$h_i(t) = \mu_i + \sum_{j=1}^{M} \sum_{t_n^j < t} \sum_{b=1}^{B} w_{ijb} \tilde{\phi}_b(t - t_n^j) = \mu_i + \sum_{j=1}^{M} \sum_{b=1}^{B} w_{ijb} \sum_{t_n^j < t} \tilde{\phi}_b(t - t_n^j)$$

$$= \mu_i + \sum_{j=1}^{M} \sum_{b=1}^{B} w_{ijb} \Phi_{jb}(t) = \mathbf{w}_i^T \cdot \boldsymbol{\Phi}(t), \tag{4}$$

where $\Phi_{jb}(t)$ is the convolution of $j$-th dimensional observation with $b$-th basis function and can be precomputed; $\mathbf{w}_i = [\mu_i, w_{i11}, \ldots, w_{iMB}]^T$ and $\boldsymbol{\Phi}(t) = [1, \Phi_{11}(t), \ldots, \Phi_{MB}(t)]^T$, both are $(MB + 1) \times 1$ vectors. A similar model is used in Linderman (2016) where a binary variable is included to characterize the sparsity of functional connection. As shown later, the sparsity in our model is guaranteed by utilizing a Laplace prior on weight instead.

In this paper, the basis functions are scaled (shifted) Beta densities, but alternatives such as Gaussian or Gamma also can be used. The reason we choose Beta distribution is the inference of weights will be subject to edge effects with infinite support densities when close to the endpoints of $[0, T_\phi]$. The weighted sum of Beta densities is a natural choice. With appropriate mixing, it can be used to approximate functions on bounded intervals arbitrarily well (Kottas, 2006).

## 3 Inference

The likelihood of a point process model is provided in Daley & Vere-Jones (2003). Correspondingly, the probability density (likelihood) of SNMHP on the $i$-th dimension as a function of parameters in continuous time is

$$p(D|\mathbf{w}_i, \overline{\lambda}_i) = \prod_{n=1}^{N_i} \overline{\lambda}_i \sigma(h_i(t_n^i)) \exp\left(-\int_0^T \overline{\lambda}_i \sigma(h_i(t)) dt\right). \tag{5}$$

It is worth noting that $h_i(t)$ depends on $\mathbf{w}_i$ and observations on all dimensions. Our goal is to infer the parameters i.e., weights and intensity upper-bounds, from observations, e.g., neural spike trains,

over a time interval $[0, T]$. The functional connectivity in cortical circuits is demonstrated to be sparse in neuroscience (Thomson & Bannister, 2003; Sjöström et al., 2001). To include sparsity, a factorizing Laplace prior is applied on the weights which characterize the functional connection. With the likelihood Eq. 5 and Laplace prior $p_{\mathrm{L}}(\mathbf{w}_i) = \prod_{j,b} \frac{1}{2\alpha} \exp\left(-\frac{|w_{ijb}|}{\alpha}\right)$, the log-posterior corresponds to a L1 penalized log-likelihood. The $i$-th dimensional MAP estimate can be expressed as

$$\mathbf{w}_i^*, \overline{\lambda}_i^* = \operatorname{argmax}\left\{\log p(D|\mathbf{w}_i, \overline{\lambda}_i) + \log p_L(\mathbf{w}_i)\right\}, \tag{6}$$

where $\mathbf{w}_i^*$ and $\overline{\lambda}_i^*$ are MAP estimates. The dependency of the log-posterior on parameters is complex because the sigmoid function exists in the log-likelihood term and the absolute value function exists in the log-prior term. As a result, we have no closed-form solutions for the MAP estimates. Numerical optimization methods can be applied, but unfortunately, the efficiency is low due to the high dimensionality of parameters which is $(MB + 2) \times M$. To circumvent this issue, three sets of auxiliary latent variables: Pólya-Gamma variables, latent marked Poisson processes and sparsity variables are augmented to make the weights appear in a Gaussian form in the posterior. As a result, an efficient EM algorithm with analytical updates is derived to obtain the MAP estimate.

## 3.1 Augmentation of Pólya-Gamma Variables

Following Polson et al. (2013), the binomial likelihoods parametrized by log odds can be represented as mixtures of Gaussians w.r.t. a Pólya-Gamma distribution. Therefore, we can define a Gaussian representation of the sigmoid function

$$\sigma(z) = \int_0^\infty e^{f(\omega,z)} p_{\mathrm{PG}}(\omega|1,0) d\omega, \tag{7}$$

where $f(\omega, z) = z/2 - z^2\omega/2 - \log 2$ and $p_{\mathrm{PG}}(\omega|1,0)$ is the Pólya-Gamma distribution with $\omega \in \mathbb{R}^+$. Substituting Eq. 7 into the likelihood Eq. 5, the products of observations $\sigma(h_i(t_n^i))$ are transformed into a Gaussian form.

## 3.2 Augmentation of Marked Poisson Processes

Inspired by Donner & Opper (2018), a latent marked Poisson process is augmented to linearize the exponential integral term in the likelihood. Applying the property of sigmoid function $\sigma(z) = 1 - \sigma(-z)$ and Eq.7, the exponential integral term is transformed to

$$\exp\left(-\int_0^T \overline{\lambda}_i \sigma(h_i(t)) dt\right) = \exp\left(-\int_0^T \int_0^\infty \left(1 - e^{f(\omega, -h_i(t))}\right) \overline{\lambda}_i p_{\mathrm{PG}}(\omega|1,0) d\omega dt\right). \tag{8}$$

The right hand side is a characteristic functional of a marked Poisson process. According to the Campbell's theorem (Kingman, 2005) (App. I), the exponential integral term can be rewritten as

$$\exp\left(-\int_0^T \overline{\lambda}_i \sigma(h_i(t)) dt\right) = \mathbb{E}_{p_{\lambda_i}}\left[\prod_{(\omega,t)\in\Pi_i} e^{f(\omega, -h_i(t))}\right], \tag{9}$$

where $\Pi_i = \{(\omega_k^i, t_k^i)\}_{k=1}^{K_i}$ denotes a realization of a marked Poisson process and $p_{\lambda_i}$ is the probability measure of the marked Poisson process $\Pi_i$ with intensity $\lambda_i(t, \omega) = \overline{\lambda}_i p_{\mathrm{PG}}(\omega|1,0)$. The events $\{t_k^i\}_{k=1}^{K_i}$ follow a Poisson process with rate $\overline{\lambda}_i$ and the latent Pólya-Gamma variable $\omega_k^i$ denotes the independent mark at each location $t_k^i$. We can see that, after substituting Eq. 9 into the likelihood Eq. 5, the exponential integral term is also transformed into a Gaussian form.

## 3.3 Augmentation of Sparsity Variables

The augmentation of two auxiliary latent variables above makes the augmented likelihood become a Gaussian form w.r.t. the weights. However, the absolute value in the exponent of the Laplace prior hampers the Gaussian form of weights in the posterior. To circumvent this issue, we augment the third set of auxiliary latent variables: sparsity variables. It has been proved that a Laplace

distribution can be represented as an infinite mixture of Gaussians (Donner & Opper, 2017; Pontil et al., 2000)

$$p_{\mathrm{L}}(w_{ijb}) = \frac{1}{2\alpha} \exp\left(-\frac{|w_{ijb}|}{\alpha}\right) = \int_0^\infty \sqrt{\frac{\beta_{ijb}}{2\pi\alpha^2}} \exp\left(-\frac{\beta_{ijb}}{2\alpha^2}w_{ijb}^2\right) p(\beta_{ijb}) d\beta_{ijb}, \qquad (10)$$

where $p(\beta_{ijb}) = (\beta_{ijb}/2)^{-2} \exp\left(-1/(2\beta_{ijb})\right)$. It is straightforward to see the weights are transformed into a Gaussian form in the prior after the augmentation of latent sparsity variables $\beta$.

### 3.4 Augmented Likelihood and Prior

After the augmentation of three sets of latent variables, we obtain the augmented joint likelihood and prior (derivation in App. II)

$$p(D, \Pi_i, \boldsymbol{\omega}_i|\mathbf{w}_i, \overline{\lambda}_i) = \prod_{n=1}^{N_i} \left[\lambda_i(t_n^i, \omega_n^i)e^{f(\omega_n^i, h_i(t_n^i))}\right] \cdot p_{\lambda_i}(\Pi_i|\overline{\lambda}_i) \prod_{(\omega,t)\in\Pi_i} e^{f(\omega,-h_i(t))}, \qquad (11a)$$

$$p(\mathbf{w}_i, \boldsymbol{\beta}_i) = \prod_{j,b}^{MB+1} \sqrt{\frac{\beta_{ijb}}{2\pi\alpha^2}} \exp\left(-\frac{\beta_{ijb}}{2\alpha^2}w_{ijb}^2\right) \left(\frac{2}{\beta_{ijb}}\right)^2 \exp\left(-\frac{1}{2\beta_{ijb}}\right), \qquad (11b)$$

where $\boldsymbol{\omega}_i$ is the vector of $\omega_n^i$ on each $t_n^i$, $\boldsymbol{\beta}_i$ is a $(MB+1) \times 1$ vector of $[\beta_{i00}, \beta_{i11}, \ldots, \beta_{iMB}]^T$, $\lambda_i(t_n^i, \omega_n^i) = \overline{\lambda}_i p_{\mathrm{PG}}(\omega_n^i|1, 0)$. The motivation of augmenting auxiliary latent variables should now be clear: the augmented likelihood and prior contain the weights in a Gaussian form, which corresponds to a quadratic expression for the log-posterior (L1 penalized log-likelihood).

### 3.5 EM Algorithm

The original MAP estimate has been represented by Eq. 6. With the support of auxiliary latent variables, we propose an analytical EM algorithm to obtain the MAP estimate instead of performing numerical optimization. In the standard EM algorithm framework, the lower-bound (surrogate function) of the log-posterior can be represented as

$$\mathcal{Q}(\mathbf{w}_i, \overline{\lambda}_i|\mathbf{w}_i^{s-1}, \overline{\lambda}_i^{s-1}) = \mathbb{E}_{\Pi_i, \boldsymbol{\omega}_i}\left[\log p(D, \Pi_i, \boldsymbol{\omega}_i|\mathbf{w}_i, \overline{\lambda}_i)\right] + \mathbb{E}_{\boldsymbol{\beta}_i}\left[\log p(\mathbf{w}_i, \boldsymbol{\beta}_i)\right], \qquad (12)$$

with expectation over posterior distributions $p(\Pi_i, \boldsymbol{\omega}_i|\mathbf{w}_i^{s-1}, \overline{\lambda}_i^{s-1})$ and $p(\boldsymbol{\beta}_i|\mathbf{w}_i^{s-1}, \overline{\lambda}_i^{s-1})$, $s-1$ indicating parameters from last iteration.

**E step**: Based on joint distributions in Eq. 11, the posterior of latent variables can be derived. The detailed derivation is provided in App. III. The posterior distributions of Pólya-Gamma variables $\boldsymbol{\omega}_i$ and sparsity variables $\boldsymbol{\beta}_i$, and the posterior intensity of marked Poisson process $\Pi_i$ are

$$p(\boldsymbol{\omega}_i|\mathbf{w}_i^{s-1}) = \prod_{n=1}^{N_i} p_{\mathrm{PG}}(\omega_n^i|1, h_i^{s-1}(t_n^i)), \qquad (13a)$$

$$\Lambda_i(t, \omega|\mathbf{w}_i^{s-1}, \overline{\lambda}_i^{s-1}) = \overline{\lambda}_i^{s-1}\sigma(-h_i^{s-1}(t))p_{\mathrm{PG}}(\omega|1, h_i^{s-1}(t)), \qquad (13b)$$

$$p(\boldsymbol{\beta}_i|\mathbf{w}_i^{s-1}) = \prod_{j,b}^{MB+1} p_{\mathrm{IG}}(\beta_{ijb}|\frac{\alpha}{w_{ijb}^{s-1}}, 1), \qquad (13c)$$

where $\Lambda_i(t, \omega)$ is the posterior intensity of $\Pi_i$, $p_{\mathrm{IG}}$ is the inverse Gaussian distribution. It is worth noting that $h_i^{s-1}(t)$ depends on $\mathbf{w}_i^{s-1}$. The first order moments, $\mathbb{E}[\omega_n^i] = 1/(2h_i^{s-1}(t_n^i))\tanh(h_i^{s-1}(t_n^i)/2)$ and $\mathbb{E}[\beta_{ijb}] = \alpha/w_{ijb}^{s-1}$, will be used in the M step.

**M step**: Substituting Eq. 13 into Eq. 12, we obtain the lower-bound $\mathcal{Q}(\mathbf{w}_i, \overline{\lambda}_i|\mathbf{w}_i^{s-1}, \overline{\lambda}_i^{s-1})$. The updated parameters can be obtained by maximizing the lower-bound. The detailed derivation is provided in App. III. Due to the augmentation of auxiliary latent variables, the update of parameters has a closed-form solution

$$\overline{\lambda}_i^s = (N_i + K_i)/T, \qquad (14a)$$

$$\mathbf{w}_i^s = \boldsymbol{\Sigma}_i \int_0^T B_i(t)\boldsymbol{\Phi}(t)dt, \qquad (14b)$$

where $K_i = \int_0^T \int_0^\infty \Lambda_i(t, \omega | \mathbf{w}_i^{s-1}, \overline{\lambda}_i^{s-1}) d\omega dt$, $\mathbf{\Sigma}_i = \left[ \int_0^T A_i(t) \mathbf{\Phi}(t) \mathbf{\Phi}^T(t) dt + \text{diag}\left( \alpha^{-2} \mathbb{E}[\boldsymbol{\beta}_i] \right) \right]^{-1}$
with $\text{diag}(\cdot)$ indicating the diagonal matrix of a vector, $A_i(t) = \sum_{n=1}^{N_i} \mathbb{E}[\omega_n^i] \delta(t - t_n^i) + \int_0^\infty \omega \Lambda_i(t, \omega) d\omega$, $B_i(t) = \frac{1}{2} \sum_{n=1}^{N_i} \delta(t - t_n^i) - \frac{1}{2} \int_0^\infty \Lambda_i(t, \omega) d\omega$ with $\delta(\cdot)$ being the Dirac delta function. It is worth noting that numerical quadrature methods, e.g., Gaussian quadrature, need to be applied to intractable integrals above.

## 3.6 COMPLEXITY AND HYPERPARAMETERS

The complexity of our proposed EM algorithm is $\mathcal{O}(N N_{T_\phi} B + L(N(MB+1)^2 + M(MB+1)^3))$ where $N$ is the number of observations on all dimensions, $N_{T_\phi}$ is the the average number of observations on the support of $T_\phi$ on all dimensions and $L$ is the number of iterations. The first term is due to the convolution nature of Hawkes process, the second and third term to the matrix multiplication and inversion in EM iterations. For one application, the number of dimensions $M$ and basis functions $B$ are fixed and much less than $N$. Therefore, the complexity can be simplified as $\mathcal{O}(N(N_{T_\phi} + L))$.

The hyperparameter $\alpha$ in Laplace prior that encodes the sparsity of weights and parameters of basis functions can be chosen by cross validation or maximizing the lower-bound $\mathcal{Q}$ using numerical methods. For the number of basis functions: in essence, a large number leads to a

---

**Algorithm 1:** EM inference for SNMHP

**Result:** $\{\lambda_i(t) = \overline{\lambda}_i \sigma(\mathbf{w}_i^T \cdot \mathbf{\Phi}(t))\}_{i=1}^M$
Predefine basis functions $\{\tilde{\phi}_b(\cdot)\}_{b=1}^B$;
Initialize the hyperparameter $\alpha$ and $\{\overline{\lambda}_i, \mathbf{w}_i, \boldsymbol{\omega}_i, \Pi_i, \boldsymbol{\beta}_i\}_{i=1}^M$;
**for** *Iteration* **do**
    **for** *Dimension $i$* **do**
        Update the posterior of $\boldsymbol{\omega}_i$ by Eq. 13a;
        Update the posterior intensity of $\Pi_i$ by Eq. 13b;
        Update the posterior of $\boldsymbol{\beta}_i$ by Eq. 13c;
        Update the intensity upper-bound $\overline{\lambda}_i$ by Eq. 14a;
        Update the weights $\mathbf{w}_i$ by Eq. 14b.
    **end**
    Update the hyperparameter $\alpha$.
**end**

---

more flexible functional space while a small number results in a faster inference. In experiments, we gradually increase it until no more significant improvement. Similarly, the number of quadrature nodes and EM iterations is also gradually increased until a suitable value. The pseudocode is provided in Alg. 1.

## 4 EXPERIMENTS

We validate the EM algorithm for SNMHP in analyzing both synthetic and real-world spike data collected from the cat primary visual cortex. For comparison, the following most relevant baselines are considered: **(1)** *parametric linear multivariate Hawkes processes* that are vanilla multivariate Hawkes processes with exponential decay influence functions, for which the inference is performed by MLE (Ozaki, 1979); **(2)** *nonparametric linear multivariate Hawkes processes* with flexible influence functions, for which the inference is by majorization minimization Euler-Lagrange (MMEL) (Zhou et al., 2013); **(3)** *parametric nonlinear multivariate Hawkes processes* with exponential decay influence functions, for which the inference is by MCMC based on augmentation and Poisson thinning (MCMC-Aug) (Apostolopoulou et al., 2019). The implementation of our model is publicly available at https://github.com/zhoufeng6288/SNMHawkesBeta.

## 4.1 SYNTHETIC DATA

We analyze spike trains obtained from the synthetic network model shown in Fig. 1a. The synthetic neural network contains four groups of two neurons each. In each group, the 2 neurons are self-exciting and mutual-inhibitive while groups are independent of each other. We assume 4 scaled (shifted) Beta distributions as basis functions with support $[0, T_\phi = 6]$ in Fig. 1b. For the ground truth, it is assumed that $\phi_{11} = \phi_{33} = \phi_{55} = \phi_{77} = \tilde{\phi}_1$, $\phi_{22} = \phi_{44} = \phi_{66} = \phi_{88} = \tilde{\phi}_4$, $\phi_{12} = \phi_{34} = \phi_{56} = \phi_{78} = -\frac{1}{2}\tilde{\phi}_2$, $\phi_{21} = \phi_{43} = \phi_{65} = \phi_{87} = -\frac{1}{2}\tilde{\phi}_3$ with positive indicating excitation and negative indicating inhibition. With base activation $\{\mu_i\}_{i=1}^8 = 0$ and upper-bounds $\{\overline{\lambda}_i\}_{i=1}^8 = 5$, we use the thinning algorithm (Ogata, 1998) to generate two sets of synthetic spike

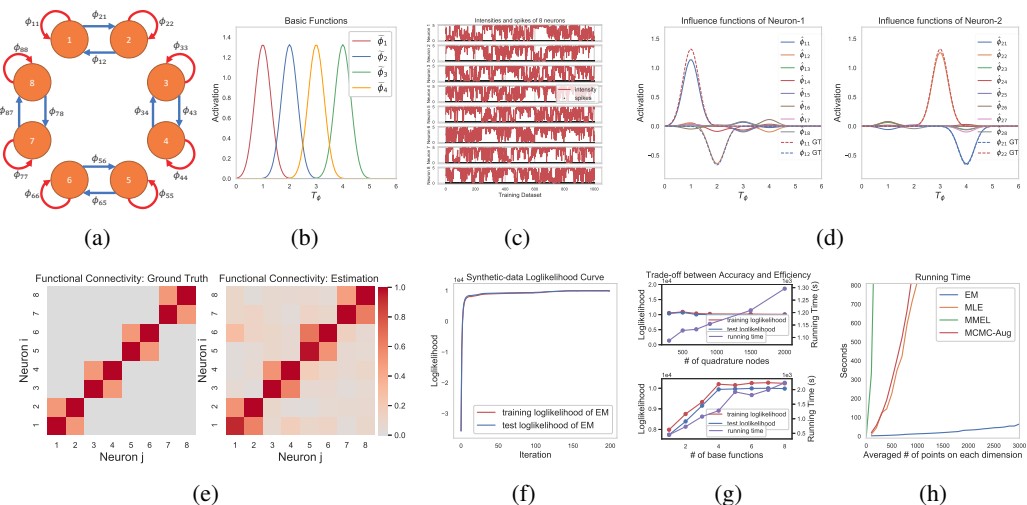

Figure 1: The synthetic network model and experimental results. (a): The synthetic neural population contains 4 independent groups. In each group, the interdependencies between 2 neurons are self-exciting and mutual-inhibitive with red arrows indicating excitation and blue arrows indicating inhibition. (b): Four scaled (shifted) Beta densities as basis functions on the support of $[0, 6]$. (c): The intensities and spike times of 8 neurons in the synthetic data. (d): The estimated influence functions of 1-st and 2-nd neurons where the estimated $\hat{\phi}_{11}, \hat{\phi}_{12}, \hat{\phi}_{21}, \hat{\phi}_{22}$ are close to the ground truth, the other ground truth $\phi_{13...18}$ and $\phi_{23...28}$ are not labeled since they are all zero (GT=Ground Truth). (e): The heat map of functional connectivity among neural population with ground truth (left) and estimation (right). (f): The training and test log-likelihood curve w.r.t. EM iterations. (g): The trade-off between accuracy and efficiency w.r.t. # of quadrature nodes and basis functions for synthetic data. (h): The running time of 2D data for EM algorithm and alternatives w.r.t. the average observation number on each dimension (the precomputation of $\mathbf{\Phi}(t)$ is included).

data on the time window $[0, T = 1000]$ with one being the training dataset in Fig. 1c and the other one test dataset in App. IV. Each dataset contains 8 sequences and each sequence consists of 3340 events on average. We aim to identify the functional connectivity of the neural population and the temporal dynamics of influence functions from statistically dependent spike trains. More experimental details, e.g., hyperparameters, are given in the App. IV.

The temporal dynamics of interactions among the neural population is shown in Fig. 1d where we plot the estimated influence functions of 1-st and 2-nd neurons (other neurons are shown in the App. IV). The estimated $\hat{\phi}_{11}$ and $\hat{\phi}_{22}$ exhibit the self-exciting relation with $\hat{\phi}_{12}$ and $\hat{\phi}_{21}$ characterizing the mutual-inhibitive interactions. All estimated influence functions are in a flexible form and close to the ground truth. Besides, as shown in Fig. 1e, the estimated functional connectivity recovers the ground-truth structure successfully. The functional connectivity is defined as $\int |\phi_{ij}(t)| dt$ meaning there is no connection only if neither excitation nor inhibition exists.

The training and test log-likelihood (LogL) curves w.r.t. EM iterations are shown in Fig. 1f where our EM algorithm converges fast with only 50 iterations needed to obtain a plateau. The trade-off between accuracy (LogL) and efficiency (running time) w.r.t. the number of quadrature nodes and basis functions is shown in Fig. 1g where we can see the accuracy is not sensitive to

Table 1: Training/test LogL ($\times 10^3$) of different models for synthetic data.

|  | MLE | MMEL | MCMC-Aug | EM |
|---|---|---|---|---|
| Training LogL | 2.051 | 1.993 | 2.199 | **2.465** |
| Test LogL | 1.866 | 1.843 | 2.278 | **2.373** |

the number of quadrature nodes over 100 and the optimal number of basis functions is 4. A larger number does not significantly improve the accuracy but leads to a longer running time. Moreover, we compare the running time of our method with alternatives in Fig. 1h where the number of dimensions $M$ is fixed to 2, basis functions $B$ to 4, quadrature nodes to 200 and iterations of all methods to 200. We can observe that our EM algorithm is the most efficient, even superior to MLE for the classic parametric case, which verifies its efficiency. Also, we compare our model's fitting and pre-

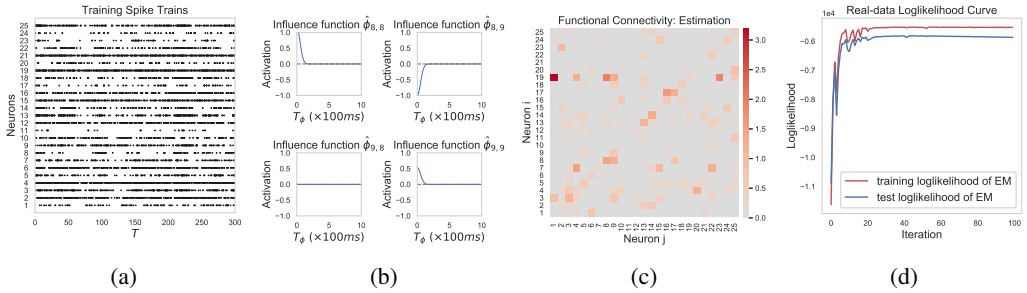

Figure 2: The real data experimental results. (a): The training spike trains extracted from real data (test spike trains in App. IV). (b): The estimated influence functions between 8-th and 9-th neurons. (c): The heat map of estimated functional connectivity among 25 neurons. (d): The training and test LogL curves w.r.t. EM iterations.

diction ability with baseline models for 1-st and 2-nd neurons. Training and test LogL are shown in Tab. 1 where our SNMHP with EM inference is the champion due to its superior generalized expressive ability.

## 4.2 REAL DATA

In this section, we analyze our model performance on a real multi-neuron spike train dataset. We aim to draw some conclusions about the functional connectivity of cortical circuits and make inferences of the temporal dynamics of influence.

**Spike Train Data** (Blanche, 2005; Apostolopoulou et al., 2019) Several multi-channel silicon electrode arrays are designed to record simultaneously spontaneous neural activity of multiple isolated single units in anesthetized paralyzed cat primary visual cortex areas 17. The spike train dataset contains spike times of 25 simultaneously recorded neurons.

**Preliminary Setup** We extract the spike times in the time window $[0, 300]$ (time unit: 100ms, the same applies to the following) as the training data (Fig. 2a) and $[300, 600]$ as the test data (App. IV). Both datasets contain approximate 7000 timestamps. All hyperparameters are fine tuned to obtain the maximum test LogL: the scaled (shifted) Beta distribution $\text{Beta}(\tilde{\alpha} = 50, \tilde{\beta} = 50, \text{shift} = -5)$ with support $[0, T_\phi = 10]$ is designed as the basis function; the number of quadrature nodes is set to 1000 and EM iterations to 100. More experimental details, e.g., hyperparameters, are given in the App. IV.

**Results** $25 \times 25$ influence functions among the neuron population are estimated in the application. An example of the influence functions between 8-th and 9-th neurons are plotted in Fig. 2b where our SNMHP model successfully captures the exciting or inhibitive interaction between neurons. Besides, the estimated functional connectivity is shown in Fig. 2c where we can see the functional connection structure among neural population is sparse. Unfortunately, because the ground-truth functional connectivity of cortical circuits is unknown, the estimated functional connectivity cannot be compared with the ground truth but here we resort to the test LogL to verify whether the estimation is good. The training and test LogL curves are shown in Fig. 2d where they both reach a close plateau indicating the estimation is appropriate without overfitting or underfitting.

A significant advantage of our EM algorithm is the efficiency. The 25-dimensional observation in the real data is a challenge for the inference. For the running time, our EM algorithm costs 3 minutes, the MCMC-Aug costs 1 hour and 45 minutes with the same number of iterations while MLE and MMEL cannot finish in 2 days due to the curse of dimensionality. Moreover, the fitting and prediction ability is compared in Tab. 2. The superior performance of SNMHP w.r.t. training and test LogL

Table 2: Training/test LogL ($\times 10^3$) and running times of different models for real data.

|  | MLE | MMEL | MCMC-Aug | EM |
|---|---|---|---|---|
| Training LogL | - | - | -15.328 | **-5.519** |
| Test LogL | - | - | -6.133 | **-5.862** |
| Running Time | > 2 days | > 2 days | 1h 45m | **3m** |

demonstrates our model can capture the complex mixture of exciting and inhibitive interactions among neural population which leads to better goodness-of-fit.

## 5 DISCUSSION AND CONCLUSION

Although we propose a point-estimation method (EM algorithm) in this work, a straightforward extension to Gibbs sampler is already at hand. Based on the augmented likelihood and prior, we can obtain the conditional densities of latent variables and parameters in *closed form*, which constitutes a Gibbs sampler with better efficiency than MCMC-Aug since the time-consuming Metropolis-Hasting sampling in MCMC-Aug is not needed. However, the proposed Gibbs sampler is less efficient than the proposed EM algorithm because the latent Poisson processes have to be sampled by thinning algorithm in Gibbs sampler which is time consuming. For the model in Apostolopoulou et al. (2019), a tighter intensity upper-bound is used to reduce the number of thinned points to accelerate the sampler. Instead, our EM algorithm does not encounter this problem as we compute the expectation rather than sampling. Moreover, Apostolopoulou et al. (2019) can only use one basis function, which limits influence functions to be purely exciting or inhibitive exponential decay. Instead, our model can utilize multiple basis functions to characterize an influence function that is a mixture of excitation and inhibition.

In this paper, we develop a SNMHP model in the continuous-time regime which can characterize excitation-inhibition-mixture temporal dependencies among the neural population. Three auxiliary latent variables are augmented to make the corresponding EM algorithm in a closed form to improve efficiency. The synthetic and real data experimental results confirm that our model's accuracy and efficiency are superior to the state of the arts. From the application perspective, although our model is proposed in the neuroscience domain, it can be applied to other applications where the inhibition is a vital factor, e.g., in the coronavirus (COVID-19) spread, the inhibitive effect may represent the medical treatment or cure, or the forced isolation by government. From the inference perspective, our EM algorithm is a point-estimation method; other efficient distribution-estimation methods can be developed, e.g., the Gibbs sampler mentioned above or the mean-field variational inference.

### ACKNOWLEDGMENTS

The authors would like to thank the anonymous reviewers for insightful comments which greatly improved the paper. This work was supported by NSFC Projects (Nos. 62061136001, 61620106010), Beijing NSF Project (No. JQ19016), Beijing Academy of Artificial Intelligence (BAAI), Tsinghua-Huawei Joint Research Program, a grant from Tsinghua Institute for Guo Qiang, Tiangong Institute for Intelligent Computing, and the NVIDIA NVAIL Program with GPU/DGX Acceleration. F. Zhou was partially funded by China Postdoctoral Science Foundation.

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

APPENDIX

## I CAMPBELL'S THEOREM

Let $\Pi_{\hat{\mathcal{Z}}} = \{(\mathbf{z}_n, \boldsymbol{\omega}_n)\}_{n=1}^N$ be a marked Poisson process on the product space $\hat{\mathcal{Z}} = \mathcal{Z} \times \Omega$ with intensity $\Lambda(\mathbf{z}, \boldsymbol{\omega}) = \Lambda(\mathbf{z})p(\boldsymbol{\omega}|\mathbf{z})$. $\Lambda(\mathbf{z})$ is the intensity for the unmarked Poisson process $\{\mathbf{z}_n\}_{n=1}^N$ with $\boldsymbol{\omega}_n \sim p(\boldsymbol{\omega}_n|\mathbf{z}_n)$ being an independent mark drawn at each $\mathbf{z}_n$. Furthermore, we define a function $h(\mathbf{z}, \boldsymbol{\omega}) : \mathcal{Z} \times \Omega \to \mathbb{R}$ and the sum $H(\Pi_{\hat{\mathcal{Z}}}) = \sum_{(\mathbf{z}, \boldsymbol{\omega}) \in \Pi_{\hat{\mathcal{Z}}}} h(\mathbf{z}, \boldsymbol{\omega})$. If $\Lambda(\mathbf{z}, \boldsymbol{\omega}) < \infty$, then

$$\mathbb{E}_{\Pi_{\hat{\mathcal{Z}}}} \left[ \exp\left(\xi H(\Pi_{\hat{\mathcal{Z}}})\right) \right] = \exp\left[ \int_{\hat{\mathcal{Z}}} \left( e^{\xi h(\mathbf{z}, \boldsymbol{\omega})} - 1 \right) \Lambda(\mathbf{z}, \boldsymbol{\omega}) d\boldsymbol{\omega} d\mathbf{z} \right],$$

for any $\xi \in \mathbb{C}$. The above equation defines the characteristic functional of a marked Poisson process. This proves Eq.9 in the main paper. The mean is

$$\mathbb{E}_{\Pi_{\hat{\mathcal{Z}}}} \left[ H(\Pi_{\hat{\mathcal{Z}}}) \right] = \int_{\hat{\mathcal{Z}}} h(\mathbf{z}, \boldsymbol{\omega}) \Lambda(\mathbf{z}, \boldsymbol{\omega}) d\boldsymbol{\omega} d\mathbf{z},$$

which is used when substituting Eq. 13 into Eq. 12.

## II DERIVATION OF AUGMENTED LIKELIHOOD AND PRIOR

Substituting Eq.7 and 9 into Eq.5 in the main paper, the augmented likelihood is obtained

$$\begin{aligned}
p(D|\mathbf{w}_i, \overline{\lambda}_i) &= \prod_{n=1}^{N_i} \overline{\lambda}_i \sigma(h_i(t_n^i)) \exp\left( -\int_0^T \overline{\lambda}_i \sigma(h_i(t)) dt \right) \\
&= \prod_{n=1}^{N_i} \left( \int_0^\infty \overline{\lambda}_i e^{f(\omega_n^i, h_i(t_n^i))} p_{\mathrm{PG}}(\omega_n^i|1, 0) d\omega_n^i \right) \cdot \mathbb{E}_{p_{\lambda_i}} \left[ \prod_{(\omega, t) \in \Pi_i} e^{f(\omega, -h_i(t))} \right] \\
&= \iint \prod_{n=1}^{N_i} \left[ \lambda_i(t_n^i, \omega_n^i) e^{f(\omega_n^i, h_i(t_n^i))} \right] \cdot p_{\lambda_i}(\Pi_i|\overline{\lambda}_i) \prod_{(\omega, t) \in \Pi_i} e^{f(\omega, -h_i(t))} d\boldsymbol{\omega}_i d\Pi_i.
\end{aligned}$$

where $\boldsymbol{\omega}_i$ is the vector of $\omega_n^i$ and $\lambda_i(t_n^i, \omega_n^i) = \overline{\lambda}_i p_{\mathrm{PG}}(\omega_n^i|1, 0)$. It is straightforward to see the augmented likelihood is

$$p(D, \Pi_i, \boldsymbol{\omega}_i|\mathbf{w}_i, \overline{\lambda}_i) = \prod_{n=1}^{N_i} \left[ \lambda_i(t_n^i, \omega_n^i) e^{f(\omega_n^i, h_i(t_n^i))} \right] \cdot p_{\lambda_i}(\Pi_i|\overline{\lambda}_i) \prod_{(\omega, t) \in \Pi_i} e^{f(\omega, -h_i(t))},$$

which is Eq.11a.

Similarly, the integrand in Eq. 10 is just the augmented prior in Eq. 11b.

## III DERIVATION OF EM ALGORITHM

In the standard EM algorithm framework, the lower-bound of log-posterior has been provided in Eq. 12. The posterior of latent variables can be derived from the joint distribution in Eq. 11. The derivation is relatively easy for $\boldsymbol{\omega}_i$ and $\boldsymbol{\beta}_i$ while $\Pi_i$ is difficult. In the following, $s-1$ and $s$ mean the last and current iteration in the EM algorithm.

### E STEP

**1.** The posterior of Pólya-Gamma variables $\boldsymbol{\omega}_i$ is dependent on the activation $h_i^{s-1}(t)$ at $\{t_n^i\}_{n=1}^{N_i}$, which is further dependent on $\mathbf{w}_i^{s-1}$ through Eq. 4

$$p(\boldsymbol{\omega}_i|\mathbf{w}_i^{s-1}) = \prod_{n=1}^{N_i} p_{\mathrm{PG}}(\omega_n^i|1, h_i^{s-1}(t_n^i)),$$

where we utilize the tilted Pólya-Gamma density $p_{\text{PG}}(\omega|b,c) \propto e^{-c^2\omega/2}p_{\text{PG}}(\omega|b,0)$ (Polson et al., 2013).

**2.** The posterior of sparsity variables $\boldsymbol{\beta}_i$ is an inverse Gaussian distribution which is dependent on weights $\mathbf{w}_i^{s-1}$

$$p(\boldsymbol{\beta}_i|\mathbf{w}_i^{s-1}) = \prod_{j,b}^{MB+1} p_{\text{IG}}(\beta_{ijb}|\frac{\alpha}{w_{ijb}^{s-1}},1).$$

**3.** The posterior of $\Pi_i$ is dependent on both $h_i^{s-1}(t)$ and $\overline{\lambda}_i^{s-1}$

$$p(\Pi_i|\mathbf{w}_i^{s-1},\overline{\lambda}_i^{s-1}) = \frac{p_{\lambda_i}(\Pi_i|\overline{\lambda}_i^{s-1})\prod_{(\omega,t)\in\Pi_i} e^{f(\omega,-h_i^{s-1}(t))}}{\int p_{\lambda_i}(\Pi_i|\overline{\lambda}_i^{s-1})\prod_{(\omega,t)\in\Pi_i} e^{f(\omega,-h_i^{s-1}(t))}d\Pi_i}.$$

The Campbell's theorem can be applied to convert the denominator, the equation above can be transformed as

$$p(\Pi_i|\mathbf{w}_i^{s-1},\overline{\lambda}_i^{s-1}) = \frac{p_{\lambda_i}(\Pi_i|\overline{\lambda}_i^{s-1})\prod_{(\omega,t)\in\Pi_i} e^{f(\omega,-h_i^{s-1}(t))}}{\exp\left(-\iint(1-e^{f(\omega,-h_i^{s-1}(t))})\overline{\lambda}_i^{s-1}p_{\text{PG}}(\omega|1,0)d\omega dt\right)}$$
$$= \prod_{(\omega,t)\in\Pi_i}\left(e^{f(\omega,-h_i^{s-1}(t))}\overline{\lambda}_i^{s-1}p_{\text{PG}}(\omega|1,0)\right)\cdot\exp\left(-\iint e^{f(\omega,-h_i^{s-1}(t))}\overline{\lambda}_i^{s-1}p_{\text{PG}}(\omega|1,0)d\omega dt\right).$$

The above posterior distribution is in the likelihood form of a marked Poisson process with intensity function

$$\Lambda_i(t,\omega|\mathbf{w}_i^{s-1},\overline{\lambda}_i^{s-1}) = e^{f(\omega,-h_i^{s-1}(t))}\overline{\lambda}_i^{s-1}p_{\text{PG}}(\omega|1,0) = \overline{\lambda}_i^{s-1}\sigma(-h_i^{s-1}(t))p_{\text{PG}}(\omega|1,h_i^{s-1}(t)).$$

M STEP

Substituting posterior distributions of latent variables into Eq. 12, we obtain the lower-bound $\mathcal{Q}$. The first term of Eq. 12 is

$$\mathbb{E}_{\Pi_i,\boldsymbol{\omega}_i}\left[\log p(D,\Pi_i,\boldsymbol{\omega}_i|\mathbf{w}_i,\overline{\lambda}_i)\right] = -\frac{1}{2}\mathbf{w}_i^T\cdot\int_0^T A_i(t)\boldsymbol{\Phi}(t)\boldsymbol{\Phi}^T(t)dt\cdot\mathbf{w}_i + \mathbf{w}_i^T\cdot\int_0^T B_i(t)\boldsymbol{\Phi}(t)dt$$
$$-\overline{\lambda}_i T + \left(N_i + \iint\Lambda_i(t,\omega)d\omega dt\right)\log\overline{\lambda}_i + C$$

where we utilize the mean rule in Campbell's theorem, $C$ is a constant and

$$A_i(t) = \sum_{n=1}^{N_i}\mathbb{E}[\omega_n^i]\delta(t-t_n^i) + \int_0^\infty \omega\Lambda_i(t,\omega)d\omega,$$
$$B_i(t) = \frac{1}{2}\sum_{n=1}^{N_i}\delta(t-t_n^i) - \frac{1}{2}\int_0^\infty\Lambda_i(t,\omega)d\omega,$$

with $\delta(\cdot)$ being the Dirac delta function and $\mathbb{E}[\omega_n^i] = 1/(2h_i^{s-1}(t_n^i))\tanh(h_i^{s-1}(t_n^i)/2)$ (Polson et al., 2013). The integral of intensity function has no closed-form solution but can be solved by numerical quadrature methods.

The second term of Eq. 12 is

$$\mathbb{E}_{\boldsymbol{\beta}_i}\left[\log p(\mathbf{w}_i,\boldsymbol{\beta}_i)\right] = -\frac{1}{2}\mathbf{w}_i^T\cdot\text{diag}\left(\frac{\mathbb{E}[\boldsymbol{\beta}_i]}{\alpha^2}\right)\cdot\mathbf{w}_i + C,$$

where $C$ is a constant, $\mathbb{E}[\boldsymbol{\beta}_i] = \{\mathbb{E}[\beta_{ijb}]\}_{jb}^{MB+1} = \{\alpha/w_{ijb}^{s-1}\}_{jb}^{MB+1}$ and $\text{diag}(\cdot)$ indicates the diagonal matrix of a vector.

The updated parameters $\overline{\lambda}_i^s$ and $\mathbf{w}_i^s$ can be obtained by setting the gradient of $\mathcal{Q}$ to zero. Due to auxiliary variables augmentation, we can see the weights are in a quadratic form in the lower-bound, which leads to an analytical expression

$$\overline{\lambda}_i^s = (N_i + K_i)/T,$$

$$\mathbf{w}_i^s = \boldsymbol{\Sigma}_i \int_0^T B_i(t)\boldsymbol{\Phi}(t)dt,$$

where $K_i = \int_0^T \int_0^\infty \Lambda_i(t,\omega|\mathbf{w}_i^{s-1}, \overline{\lambda}_i^{s-1})d\omega dt$, $\boldsymbol{\Sigma}_i = \left[\int_0^T A_i(t)\boldsymbol{\Phi}(t)\boldsymbol{\Phi}^T(t)dt + \mathrm{diag}\left(\alpha^{-2}\mathbb{E}[\boldsymbol{\beta}_i]\right)\right]^{-1}$.
It is worth noting that numerical quadrature methods need to be applied to intractable integrals above.

## IV EXPERIMENTAL DETAILS

In this appendix, we elaborate on some experimental details.

### SYNTHETIC DATA EXPERIMENTS

For the synthetic data, the intensities and spike times of our simulated training and test data are shown in Fig. 1. As shown in the experiment of log-likelihood and running time w.r.t. the number of basis functions, the optimal number of basis functions is 4, which are chosen as the ground truth: $\tilde{\phi}_{\{1,2,3,4\}} = \mathrm{Beta}(\tilde{\alpha} = 50, \tilde{\beta} = 50, \mathrm{scale} = 6, \mathrm{shift} = \{-2, -1, 0, 1\})$. By cross validation, the hyperparameter $\alpha$ is chosen to be 0.05. As shown in the experiment of log-likelihood and running time w.r.t. the number of quadrature nodes, the accuracy is not sensitive to the number of quadrature nodes over 100, so the number of quadrature nodes is set to 2000. The number of EM iterations is set to 200 which is large enough for convergence. We plot the estimated influence functions of 8 neurons in Fig. 2. For comparison, we also plot the estimated influence functions of 8 neurons from vanilla multivariate Hawkes processes using the MLE algorithm in Fig. 3 and the functional connectivity graph in Fig. 4. We can see both estimated influence functions and functional connectivity graph are far from the ground truth. This demonstrates the necessity of incorporating inhibitive interaction into the model when the Hawkes process is applied in the neuroscience domain. The running time experiment and the fitting and prediction experiment are both conducted for 2 neurons because the baseline models cannot finish in 2 days with 8 neurons because of the curse of dimensionality.

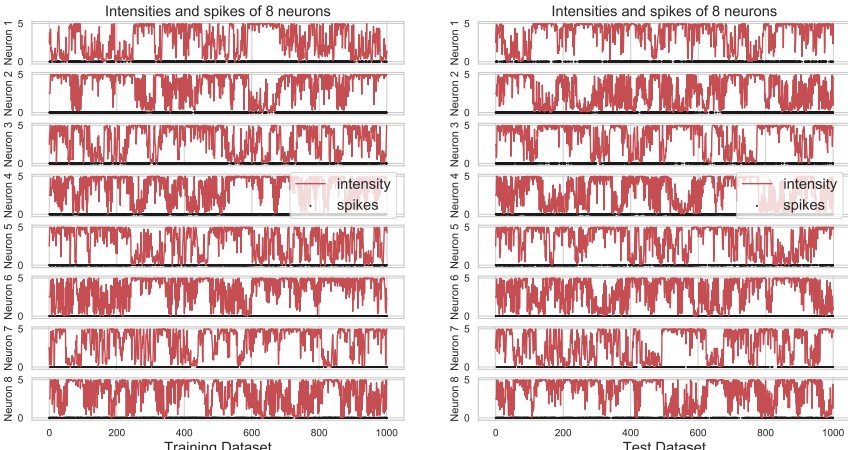

Figure 1: The intensities and spike times of 8 neurons in our synthetic training dataset (left) and test dataset (right).

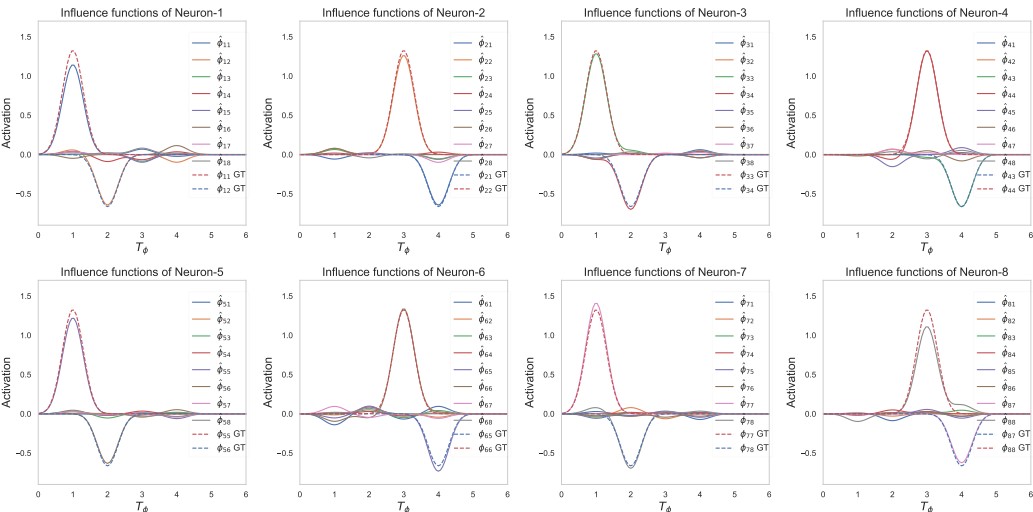

Figure 2: The estimated influence functions of all neurons where the estimated $\hat{\phi}$'s are close to the ground truth and some ground truth are not labeled since they are all zero (GT=Ground Truth).

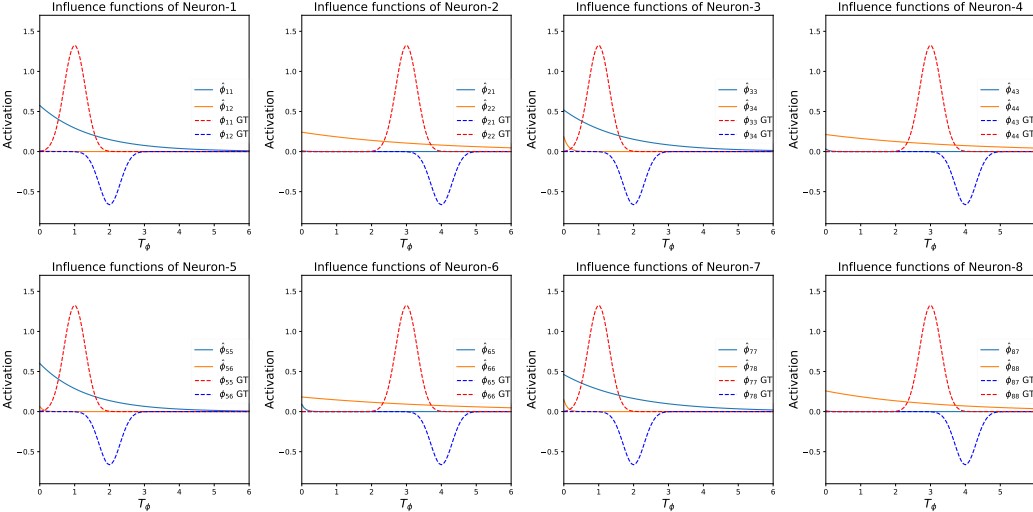

Figure 3: The estimated influence functions of all neurons from vanilla multivariate Hawkes processes using MLE; some influence functions are not labelled since they are all zero (GT=Ground Truth).

REAL DATA EXPERIMENTS

For the real spike data in cat primary visual cortex areas 17, it contains spike times of 25 simultaneously recorded neurons. We extract the spike times in the time window $[0, 300]$ (time unit: 100ms) as the training data and $[300, 600]$ as the test data. Both datasets contain approximate 7000 timestamps. The training and test spike trains are plotted in Fig. 5 below.

All hyperparameters are fine tuned in real data experiments. Specifically, the optimal basis function is chosen as: $\tilde{\phi} = \text{Beta}(\tilde{\alpha} = 50, \tilde{\beta} = 50, \text{scale} = 10, \text{shift} = -5)$. The hyperparameter $\alpha$ is optimised to be $0.1$ by cross validation. The number of quadrature nodes is chosen to be 1000 for which the running time is acceptable. The number of EM iterations is set to 100 which is large enough for convergence.

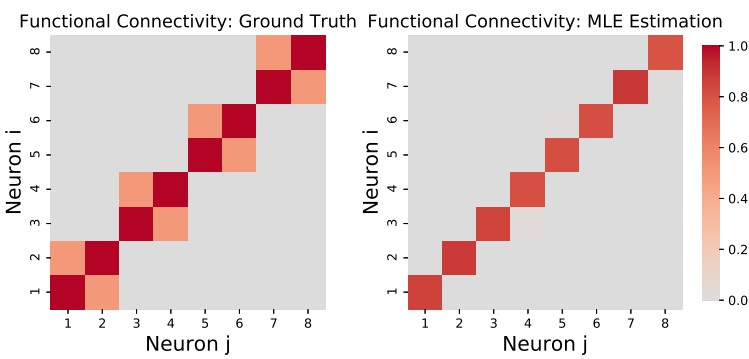

Figure 4: The heat map of functional connectivity among neural population with ground truth (left) and estimation from vanilla multivariate Hawkes processes (right).

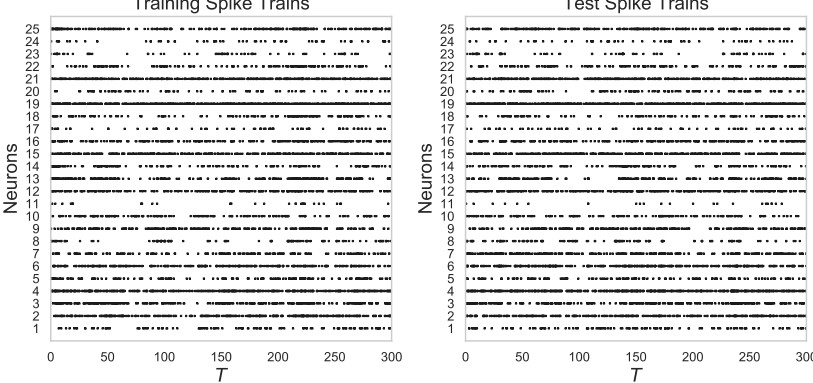

Figure 5: The training and test spike trains in the dataset of cat primary visual cortex areas 17.

