# OpenReview forum: "Efficient Inference of Flexible Interaction in Spiking-neuron Networks"
_ICLR.cc/2021/Conference — ICLR 2021 Poster_

### Official Review · AnonReviewer1 · 2020-10-26
**I think the novel method has some impacts in modeling neuron-spiking networks, and thus vote for accept.**

**Rating:** 6
**Confidence:** 3

**Review:**

This paper proposes a novel multivariate nonlinear Hawkes model by modeling the intensity as the product of a upper-bounding intensity, and a sigmoidal function that maps the real-valued latent intensities into a nonnegative real value in [0,1]. Hence, the positive exciting and negative inhibiting interactions among dimensions can be captured by the latent intensities and real-valued influence functions. To further induce sparsity over the influence networks among dimensions, the Laplace prior is used to model the weight parameters of influence functions. A standard EM algorithm is developed to perform inference. Experiments conducted on simulated data demonstrate the novel model can correctly recover the interaction network among neurons, and the influence functions, with less computation time compared with the baselines. They also compared the model with the baselines on real neuron spike data in terms of both train- and test-loglikelihood.

-Quality:  The novel Hawkes model that capture complicated interactions among dimensions, is reasonable to me. The math derivations in Sec 3, appear to be solid.

-Clarity: The motivation and structure of this paper is pretty clear. The authors clearly introduces the main techniques they used from [Adams2009,Donner2017], and their novel Hawkes model.

-Originality & Significance: Although many techniques (sigmoidal function framework, Laplace prior) have been well studied before, they are first reflected in this context of capturing both exciting and inhibiting interactions by a novel multivariate nonlinear Hawkes model. Although the new contributions seem to be incremental, it may has some impacts for modeling complicated neuron interactions. Specifically, the low computational complexity of their novel model is very clear compared with those baselines.

-Pros: A sensible and useful new Hawkes model can capture complicated interactions among dimension, outperform the previous canonical Hawkes models in terms of computation efficiency and test log-likelihood.

-Cons:
I think the experiments could be more convincing. For synthetic data, the authors only consider one example of neuron interaction networks (Fig.1 a). Although the results show their model correctly recover the ground truth, I am wondering that the authors can provide more examples in supplements, in which the functional connectivity among neurons inferred by the canonical multivariate Hawkes process can be shown to clearly demonstrate why both exciting and inhibiting need to be captured.

In addition to the train- and test- loglikelihood, can you compare these methods using some other evaluation metrics? In terms of recovering the underlying functional connectivity, does it make sense to compare the estimate functional connectivity graph with the ground truth using AUROC, PR, or F1 scores?

To my knowledge, it seems to be the first attempt to use the sigmoidal function framework to capture both exciting and inhibiting interactions among dimensions in the multivariate Hawkes model. Nevertheless, I think there also exist some other related multivariate nonlinear point processes can also capture self- and mutual-inhibiting behaviors, e.g., [1]. The authors should consider more strongly related baselines for the comparisons.

[1] The Multivariate Hawkes Process in High Dimensions: Beyond Mutual Excitation. https://arxiv.org/abs/1707.04928

---

> ### Author Response · Authors · 2020-11-17
> **Thanks for the reviewer’s positive comment.**
>
> Thanks for the reviewer’s positive comment.
>
> We provide new experimental results in the supplemental material (Fig.3 and 4 in App.) where the estimated influence functions (Fig.3) and functional connectivity graph (Fig.4) among neurons from the vanilla multivariate Hawkes processes are shown. It is straightforward to see both the estimated influence functions and functional connectivity graph are far from the ground truth. This clearly demonstrates why both exciting and inhibitive interactions need to be captured.
>
> I am not sure I catch your suggestion for the AUC ROC, PR, or F1. In my understanding, the AUC ROC, PR, or F1 scores are evaluation metrics for classification problems. Because the functional connectivity graph (fig1.e) is real-valued, all these metrics cannot be used for comparison. If I misunderstand something, please correct me.
>
> For the “The Multivariate Hawkes Process in High Dimensions: Beyond Mutual Excitation”. We politely do not agree it is an appropriate baseline for comparison. In fact, the paper is orthogonal to our work. It provided the theoretical statistical analysis about the properties of the nonlinear Hawkes process, and established a concentration inequality for second-order statistics of the nonlinear Hawkes process. There is no inference algorithm proposed in that work. If I misunderstand something, please correct me.

---

### Official Review · AnonReviewer4 · 2020-10-27
**Core is good but I have some conceptual concerns, and the paper needs to be edited for clarity**

**Rating:** 7
**Confidence:** 4

**Review:**

The paper introduces a novel variation on the multivariate Hawkes process that can be fitted efficiently and that allows for interactions to have both excitatory and inhibitory effects. An EM algorithm is provided for fitting the model to empirical data (other methods are discussed in the supplement, but I didn't look at those), and application examples are given on synthetic data and on neuronal spiking data recorded in visual cortex in cat.

The core contribution of the paper is novel and potentially very useful and relevant to the ICLR community. Unfortunately, the content is not done justice to by the presentation and the quality of the writing, which could be improved or made clearer in several passages. Moreover, a couple of conceptual issues temper my enthusiasm for this work (although they should be easily addressable). All things considered, my initial recommendation is to accept. I have three main points to raise to the attention of the authors (one of which is more of a question/suggestion), and several minor issues to highlight from the point of view of clarity.

###  Main points:

1. From a conceptual standpoint, I am confused by the way in which the neuroscience term "functional connectivity" is used. Although this term is very broadly used in different contexts, it is generally meant to convey the notion of effective or statistical connectivity as opposed to direct anatomical or synaptic connectivity. For instance (to cite an influential paper in a subfield not too distant from that of the present work), when describing its model of functional connectivity among retinal cells, Pillow (2008) writes: "Although it provides an accurate functional description of correlated spike responses, the generalized linear model does not reveal the biophysical mechanisms underlying the statistical dependencies between neurons: coupling does not necessarily imply anatomical connections between cells, but could (for example) reflect dependencies due to shared input noise". This is, indeed, what the proposed method can measure. But the paper appears to use this concept interchangeably with that of synaptic connectivity (for instance, in section 4.1, first paragraph: "We aim to identify the synaptic connectivity (functional connectivity) of the neural population…"), which is problematic. Functional and synaptic connectivity may coincide in carefully constructed synthetic examples, but this is not true in general and certainly not for empirical recordings of neuronal activity.
This not just a terminology issue, as it seems to inform at least one technical choice. Indeed, when motivating the choice of basis function sets, the author state "Although basis functions can be in any form, in order for the weights to represent synapses connection strength, basis functions are chosen to be probability densities with compact support that means they have bounded support [0, $T_\phi$] and the integral is one". But by the discussion above the proposed method doesn't tell us anything about the synaptic connectivity between the measured neurons, so the interpretation of the weights as synaptic weights is unfounded. One should rather talk of functional connectivity weights. To see this in a different way, consider the discussion of the real data fit in section 4.2: for a given pair of cells, i and j, the influence function of i on j can take on positive or negative values at different lags (this is indeed one of the main points of the method). But by the argument just discussed on synaptic weights, are we supposed to interpret this as evidence for the fact that neuron i makes multiple synapses onto neuron j (with different time courses), some of which are excitatory and some inhibitory? This makes little sense from a neurobiological/synaptic point of view, while of course still being perfectly fine from the point of view of functional connectivity. I suggest that the author rephrase all their discussions of synaptic connectivity in terms of functional connectivity.
2. The point above also leads into another question/suggestion for the authors. If you were to drop the requirement that the basis functions used in equations 3 were area-normalized, other basis sets could present themselves as natural choices (I'm bringing this up as it seems that the only justification for the area-normalization requirement was the synaptic weight interpretation, which as I've explained above I believe to be misleading). In particular, B-splines come to mind: like the pdfs used in the paper, they are positive and have finite support. But additionally they offer a very precise characterisation of the class of function they allow to model (the set of splines with the same order, defined over the same knots). This allows one to immediately get an intuitive idea of the complexity of the set of functions over which inference is performed. I don't think this is easily feasible by using, say, Beta pdfs.
3. I am not sure I understand why the proposed method is called "nonparametric". It seems to me that there is a finite set of parameters ($\mu_i$ and $w_{ijb}$), whose size is fixed from the start and over which inference is performed. The mere fact of looking for a solution over a finite-dimensional space of functions (via eq 3) does not make a model "nonparametric". In my understanding, methods are called nonparametric when they involve looking for a solution in an infinite dimensional space (for instance a space of functions constrained only by, say, some smoothness and integrability requirement; but not the finite-dimensional vector space comprising all the linear combinations of a fixed, finite set of basis functions). The authors cite Zhou et al 2013 and state that "a similar nonparametric scheme" is implemented in that paper. But if I'm reading that right, that paper also infers the shape of the basis functions ($\phi$  here, $g$ in Zhou) from data, which involves solving some differential equations during parameter estimation. I see nothing of a similar nature here. And even the authors themselves write things like "Our goal is to infer the **parameters** i.e. weights and intensity upper bounds…" (just below equation 5; emphasis mine). So either I am missing something important about the proposed method (and in that case I hope the authors could clarify that in a revised version of the manuscript), or the term "nonparametric" is a misnomer here and should be removed from the name of the method and from the paper. As a side effect, it would also make the SNNMHP acronym slightly less intimidating.

### Minor points / clarity

1. Section 2, first paragraph: "in the meantime": you probably mean "at the same time"?
2. Same paragraph: "Apparently, the nonparametric nonlinear multivariate Hawkes processes are a suitable choice for representing the temporal dynamics of mutually excitatory or inhibitory interactions and synapses connectivity of neuron networks." I am very confused by that "apparently". Do you mean something like "clearly", "evidently" or "arguably" (probably the better option)?
3. Between eqs 5 and 6: "As demonstrated in neuron science (Thomson & Bannister, 2003; Sjöström et al., 2001), the synaptic connectivity in cortical circuits is unraveled to be sparse.". This sentence needs some editing. You probably meant something else than "unraveled", and "neuron science" is probably meant to be "neuroscience". Also the structure of the sentence itself is confusing — it reads as if "neuron science" could be only one of many branches of science that could try to have something factual to say about the sparseness of cortical circuits.
4. Page 4, top: "exits" repeated twice in the sentence, is probably meant to be "exists"?
5. Beginning of section 4.1: "The synthetic neural network contains 4 groups of neurons with a couple of ones in each group". How about "The synthetic neural network contains four groups of two neurons each"?
6. Section 4.2., paragraph "Spike train data": "Several multi-channel silicon electrode arrays are designed to record simultaneously spontaneous neural activity of multiple isolated single units in anesthetized paralyzed cat primary visual cortex areas 17. It contains spike times of 25 simultaneously recorded neurons." Another sentence that needs some editing. I suppose "it" is meant to refer to a dataset, but this is never mentioned as the subject of the previous sentence is "electrode arrays".
7. Same section, paragraph "results": "…the real functional connectivity of cortical circuits is unknown…". This is another point at which the general misuse of the concept of functional connectivity surfaces. Is "the real functional connectivity" even a well defined concept?
8. Similarly, a few lines below: "most synapses of the 11th neuron are bidirectional" - I don't think this analysis allows one to conclude anything about direct synaptic contacts between the recorded neurons, given that the 25 neurons are embedded in a vast network. This is an example of a statement that would be better if rephrased in terms of functional connectivity.



### References

Pillow, J. W., Shlens, J., Paninski, L., Sher, A., Litke, A. M., Chichilnisky, E. J., & Simoncelli, E. P. (2008). Spatio-temporal correlations and visual signalling in a complete neuronal population. Nature, 454(7207), 995-999.

----

Review update after revision: my concerns have been addressed.

---

> ### Author Response · Authors · 2020-11-17
> **Thanks for the instructive comments on some concepts and suggestions.**
>
> Thanks for the reviewer’s instructive comments on some concepts and suggestions.
>
> For the main points:
> We agree that the functional connectivity is generally meant to convey the notion of statistical connectivity as opposed to direct anatomical or synaptic connectivity. We rephrase all discussions of synaptic connectivity in terms of functional connectivity. The B-splines is a good suggestion; we will consider it in the future work. We reconsidered the proposed method. As the inference is performed on finite dimensions, we correct the nonparametric to be flexible.
>
> For the minor points:
> They have been corrected or rephrased to improve the clarity.

---

> > ### Comment · AnonReviewer4 · 2020-11-19
> > **Thank you, I have updated my review**
> >
> > Thank you for the response. My concerns have been addressed and I have updated my review accordingly.

---

### Official Review · AnonReviewer3 · 2020-10-27
**Review of Efficient Inference of Nonparametric Interaction in Spiking-neuron Networks**

**Rating:** 6
**Confidence:** 3

**Review:**

The authors here present an extension the Hawkes process to incorporate negative interactions. This allows for inference of excitatory and inhibitory interactions among point-processes using the Hawkes process framework. They present a novel inference procedure for this model using three augmentations to pieces of the model 1) representing the sigmoidal nonlinearity in the model (which is needed to keep the Poisson rates positive) with a mixture of Gaussians wrt a polya-gamma distribution. 2) Using Cambell's theorem to transform the marked Poisson process into gaussian form and 3) using a mixture of gaussians to represent the laplace prior over the weights. The authors fit the model to real and simulated data.

I think this work is an interesting extension to Hawkes processes, the novel EM inference procedure is impressive, and the use on real and simulated data is laudable.

However, I think more work needs to be done comparing this model to similar models used to infer E and I connectivity neural data. Since the application here seems to be primarily neural data, I wonder if this model can infer anything that existing models cannot. The first comparison that comes to mind is the GLM. There are recent non-parametric versions (see Dowling et al. 2020) that seem to me to be inferring similar time-varying interactions from the same kind of data as this extended Hawkes process model. I would also like to see more of a discussion about how this model compares to  Apostolopoulou 2019, which, to my knowledge, also uses a Hawkes process extension to capture inhibitory and excitatory connections. The authors should more directly address why their model should be used in place of these other methods.

It is possible that the contribution here is primarily the speed of the inference procedure. If so, I think more attention should be devoted to this, and more discussion as to why this is much faster than Apostolopoulou 2019. Is it also possible also to compare to an out-of-the-box variational inference method? Some further discussion of how each method scales with number of neurons or length of time series would also be useful.

I'm primarily trying to figure out what the authors want us to take away -- a new kind of model for neuroscience, an inference improvement to Apostolopoulou 2019, or something else. I find the work an interesting extension to the existing body of literature, but a clearer outline of the primary contribution is important, especially given the fact that this seems to be primarily of interest to the neuroscience community, who will want to decide whether or not to use this model on their data.

####EDIT####
I am satisfied with the reviewers response and will raise my score to a 6.

I think the authors should take care though in speaking practically about their work. The authors highlight the difference of 'excitation-inhibition mixture" in their response, which is indeed a mathematical difference to the existing work but, at least in the context of neuroscience, I am not sure if this serves any scientific purpose. Do the authors believe this has biophysical meaning or bears a relationship to neural coding?

I do still think some mention of GLMs will be useful as context for models that also infer neuron-neuron E and I interactions.

I appreciate the authors efforts to explicitly highlight the advantages of this work as compared to Apostolopoulou 2019.

---

> ### Author Response · Authors · 2020-11-17
> **Thanks for the reviewer's comment.**
>
> Thanks for the reviewer's comment. If our understanding is right, the two major concerns are: (1) the contribution of our work and (2) comparing to two related works.
>
> For the primary contribution of our work:
>
> As we stated in the last paragraph of introduction “To address the parametric and inefficient problems in aforementioned existing works, we develop a flexible sigmoid nonlinear multivariate Hawkes processes (SNMHP) model in the continuous-time regime, (1) which can represent the flexible excitation-inhibition-mixture temporal dynamics among the  neural  population, (2) with the efficient  conjugate inference.” Therefore, the primary contribution of our work is twofold: the first one is the model can represent the flexible excitation-inhibition-mixture interaction (we expain in the following that it is different with Apostolopoulou 2019); the second one is we derive an efficient conjugate inference algorithm.
>
> Comparing to similar models:
>
> Apostolopoulou 2019 is the most related baseline for our work. We made a comprehensive comparison with it in the submission. For Apostolopoulou 2019, as we stated in the third paragraph of introduction, the influence function in their model is a **parametric function (exponential decay)** that means it can only infer a **purely exciting or purely inhibitive exponential decay function** depending on the sign of excitation parameter of influence function. On the contrary, the influence function in our model is **flexible and “excitation-inhibition-mixture”** taking on positive or negative values at different lags, e.g. fig2(c). **Please note the “excitation-inhibition-mixture” is different from “excitation-or-inhibition”**. Besides, in Apostolopoulou 2019, due to the nonconjugacy of the excitation parameter of influence function, a Metropolis-Hastings sampling step has to be embedded into the Gibbs sampler making their MCMC inference inefficient. On the contrary, due to the augmentation of latent variables, our inference method (EM algorithm) is totally conjugate and has closed-form solution. In conclusion, our model is **more flexible** and our inference is **more efficient** than Apostolopoulou 2019. A more detailed discussion about the difference between our model and Apostolopoulou 2019 and why our method is faster is added in the discussion part in the end. The experiments about the fitting performance and running time is already provided in Tab. 1,2 and Fig. 1(h) (MCMC-Aug stands for Apostolopoulou 2019).
>
> For Dowling et al. 2020, this paper is not published yet as far as we know. In fact, we politely do not agree it is an appropriate baseline for comparison. The major concern is that they used an autoregressive point process model not a mutually regressive one (In essence, both our work and Apostolopoulou 2019 are mutually regressive point processes). This makes their model different from both ours and Apostolopoulou 2019. Due to the usage of autoregressive point process, there exists only $M$ influence functions for $M$ neurons (all influence functions from i-th neuron to other neurons are assumed to be the same $f_i$) in their model. This strong assumption limits their model’s fitting capability. On the contrary, there exists $M^2$ influence functions in both our work and Apostolopoulou 2019 (the influence function from i-th neuron to j-th is $f_{ij}$). This is the main reason why we think Apostolopoulou 2019 is a more appropriate baseline than Dowling et al. 2020. The GP based model in Dowling et al. 2020 can obtain the nonparametric time-varying interactions as flexible as ours, but the computation complexity is high because of the use of GP (matrix inversion for each $f_i$). Even though the sparse GP is used in that work for acceleration, the method is still slow. Besides, to maximize the ELBO in their variational inference, the optimization is performed on a high-dimensional space (mean vector, covariance matrix and hyperparameters), which is also time-consuming.  On the contrary, our EM algorithm has a closed-form solution without the need of optimization, which leads to a fast inference. If we misunderstand something, please correct us.
>
> Thanks the reviewer for raising two major concerns, as they guide us to provide a deeper discussion on the comparison with related SOTA works. We hope this correctly addresses the reviewer’s concerns, but if not, please kindly let us know ahead of time before the end of the discussion period.

---

### Official Review · AnonReviewer2 · 2020-10-30
**The authors report progress in estimating functional interactions between neurons. ICLR does publish a few papers in the area of neural data analysis. This is a lukewarm endorsement for publication.**

**Rating:** 6
**Confidence:** 3

**Review:**

The manuscript 'EFFICIENT INFERENCE OF NONPARAMETRIC INTERACTION IN SPIKING-NEURON NETWORKS' develops an expectation-maximization (EM) algorithm, based on an existing trick of introducing additional  latent variables, to obtain the maximum a posteriori (MAP) estimate of parameters of the model. In addition, the authors use a basis function set to express the influence function. Estimating functional interaction among neurons is an important practical problem.


Pros:
They seem to achieve great runtime performance compared to some existing methods, one of which involves MCMC.
The basis set allows the authors to express more complex influence functions.
The writing is clear.

Cons:
This is a special application, useful for analyzing neural data.
The work is somewhat incremental, although practically useful.
Not sure that the work provides any insight for a broader class of machine learning problem.

Smaller issues:
Figure 1 subplots are too small, with (d) and (g) particularly hard to see.

It might help to explain how to read log likelihood in table 1 and, especially, table 2. Is it obvious that we can compare the columns to each other?

---

> ### Author Response · Authors · 2020-11-17
> **Although our model is proposed in neuroscience, it can be applied to other applications where the inhibition is a vital factor.**
>
> Thanks for the reviewer’s positive comment.
>
> We would like to emphasize that, as we stated in the paper, “For the application, although our model is proposed in neuroscience, it can be applied to other applications where the inhibition is a vital factor.” In fact, the Hawkes process has been applied widely in finance [1], epidemics [2] and crime [3]; for example, in the coronavirus (COVID-19) spread, the inhibitive effect may represent the medical treatment or cure, or the forced isolation by government. We added further discussion in the conclusion in the end.
>
> Yes, the columns can be compared to each other directly in each table, as they are the fitting performance of different models on the same dataset.
>
> [1] Hawkes processes in finance, Bacry et al., 2015
>
> [2] SIR-Hawkes: Linking Epidemic Models and Hawkes Processes to Model Diffusions in Finite Populations, Rizoiu et al. 2017
>
> [3] Modeling and estimation of multi-source clustering in crime and security data, Mohler 2013.

---

### Decision · Program_Chairs · 2021-01-07
**Final Decision**

**Decision:**

Accept (Poster)

**Comment:**

This article proposes latent variable augmentation scheme for inference in nonlinear multivariate Hawkes processes. It combines existing approaches (Polya-gamma and sparsity-inducing variables) in a sensible way and is clearly written. Concerns were raised with respect to the comparison to alternative baselines, and answered by the authors. As a result, some reviewers have increased their score, and I recommend acceptance.